# Go with the Flow: the distribution of information processing in multi-path networks

## Abstract

The architectures of convolution neural networks (CNN) have a great impact on the predictive performance and efficiency of the model. Yet, the development of these architectures is still driven by trial and error, making the design of novel models a costly endeavor. To move towards a more guided process, the impact of design decisions on information processing must be understood better. This work contributes by analyzing the processing of the information in neural architectures with parallel pathways. Using logistic regression probes and similarity indices, we characterize the role of different pathways in the network during the inference process. In detail, we find that similar sized pathways advance the solution quality at a similar pace, with high redundancy. On the other hand, shorter pathways dominate longer ones by majorly transporting (and improving) the main signal, while longer pathways do not advance the solution quality directly. Additionally, we explore the situation in which networks start to "skip" layers and how the skipping of layers is expressed.

## 1 Introduction

The architectures of Convolutional Neural Network (CNN) classifiers are an important influencing factor regarding their predictive performance and computational efficiency, with many designs being proposed over the years (Simonyan & Zisserman, 2015; He et al., 2016; Szegedy et al., 2015; Tan & Le, 2019; Sandler et al., 2018). These architectures can be generally subdivided into two distinct categories. First, the sequential architectures like VGG16 by Simonyan & Zisserman (2015), which essentially consist of a sequence of layers leading from the input to the output of the network. While these architectures are simplistic in structure, they have been surpassed by multi-path architectures in efficiency and predictive performance.

Multipath architectures can be described as a directed acyclic graph, with the nodes representing the layers. Information provided at the input of a multipath network can be processed by different sequences of layers that are intertwined in the overall neural architecture. The Inception-family of networks is an example for such an architecture (Szegedy et al., 2015; 2016; 2017), where each building block features a different set of layers and allows for the parallel extraction of heterogeneous features. Networks featuring skip-connections like ResNet, MobileNetV2, MobileNetV3 and EfficientNet (He et al., 2016; Sandler et al., 2018; Howard et al., 2019; Tan & Le, 2019) are a special case of multipath architectures, since they may only feature a single sequence of layers.

However, since the skip-connections effectively allow signals to skip layers, it is possible for any signal from the input to take multiple paths to the output, thus making these architectures multi-path architectures. On the other hand, multi-path architectures require the model to implicitly make *decisions* regarding the distribution of the inference process, since some routes connecting two layers may differ in their capacity (number of parameters), depth and receptive field size. In essence, during training, the model will learn to utilize the different pathways, implicitly *deciding* how the inference process is distributed in branching paths and whether to skip certain layers.

These decisions have been discussed theoretically by authors like He et al. (2016), who for example elaborate the possibility of identity-mappings in skip-connections, which effectively enable the network to remove layers from the qualitative inference process. In this work, we empirically investigate how parallel pathways and skip connections are utilized in multi-path CNN architectures.

Our contributions can be summarized as follows:

- We find that networks with skip connections reliably skip layers when a mismatch between receptive field and input resolution occurs. This finding is in agreement with the findings of Richter et al. (2021b) and Richter et al. (2021a).

- We show that networks with parallel pathways of strongly different depth will prefer the shorter pathway over the longer pathway due to a partial vanishing gradient.

- We demonstrate that multi-path architectures with parallel pathways of roughly similar size advance the solution quality in a similar pace and with a high degree of redundancy.

## 2 RELATED WORK

### 2.1 MULTI-PATH NETWORKS

Since this work is focussed on multi-path architectures, we will introduce the most significant works regarding this type of neural architecture as well as the contemporary reasoning for the introduction of the proposed neural architectures. After the advent of CNNs, there were several works that focused on improving their performance by introducing novel architectures (Krizhevsky et al., 2012; Simonyan & Zisserman, 2015). Although one of the most intuitive ways to improve the accuracy of CNNs is by increasing the number of layers (depth), there is an upper limit to it. After a specific level, there are certain hurdles that occur such as network overfitting, vanishing gradient (Hochreiter, 1998; Dong et al., 2015), and also it becomes computationally expensive to train them. Therefore, to train such deep networks, several works proposed the idea of multipath networks (He et al., 2016; Larsson et al., 2017; Huang et al., 2018; Kuen et al., 2017). The multipath networks presented in (Mao et al., 2016; Tong et al., 2017; Srivastava et al., 2015) introduced shortcuts in the networks structure that facilitate a customized flow of information such that the vanishing gradient problem is reduced. The work by Szegedy et al. (2015) introduced sparsely connected architectures motivated by Hebbian principle to solve the problem of overfitting and then clustering them to dense matrices (Ümt V. Çatalyürek et al., 2010) so that the network can be trained efficiently on the hardware. Further improvements to this network were introduced (Szegedy et al., 2016; 2017) to increase its accuracy and training efficiency. Since the information available to the shallower layers is not available to the deeper layers, He et al. (2016) introduced the Residual Networks (ResNets) which used skip connections to preserve the gradient. The skip connections are operated by vector addition whereas in the method proposed by Huang et al. (2018), the skip connections are operated by concatenation of the feature maps.

### 2.2 RELEVANT ANALYSIS TOOLS AND SIMILARITY METRICS

Since we are interested in the information processing of CNNs, a set of analysis tools is required that allows us to analyze the processing within the neural architecture. Our main analysis tool that is used extensively throughout this work are logistic regression probes (LRP) by Alain et al. (2020). LRP are logistic regressions trained on the output of a hidden layer. Since classifiers implicitly maximize the linear separability of the data, the predictive performance of the logistic regression models on the test set allows us to track the progress of the intermediate solution quality while the data is propagated from layer to layer (see Fig. 1). The usefulness of LRP has been demonstrated by the works of Richter et al. (2020), Richter et al. (2021a) and Richter et al. (2021b), which utilize LRP to identify inefficiencies in neural architectures caused by layers not contributing qualitatively to the inference process. Alain et al. (2020) find by experimenting on multilayer perceptrons with skip connections that the skipping of a sequence of layers is observable by logistic LRP by a degradation in performance. A reproduction of this experiment on MNIST using a similar architecture can be seen in Fig. 2. In section 3 we furthermore utilize saturation proposed by Richter et al. (2020) as an additional tool for analyzing the output of hidden layers. Saturation uses PCA to approximate the subspace the data is processed, and can be interpreted as the percentage of dimensions utilized by the data in the output of space of a given layer. Thereby, highly saturated layers can be usually seen as *active*, in the sense that they contribute qualitatively to the prediction, while layers that are low saturated relative to the rest of the network tend to be unproductive. A sequence of unproductive layers is referred to as a *tail pattern*.

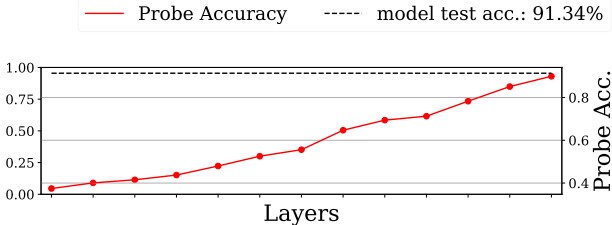

Figure 1: Logistic Regression Probes allow the practitioner to observe the evolution of the intermediate solution quality from layer to layer. The probe performance at each layer of VGG16 trained on Cifar10. The performance is increasing layer by layer, indicating that the problem is solved incrementally and that the inference process is evenly distributed among layers.
.

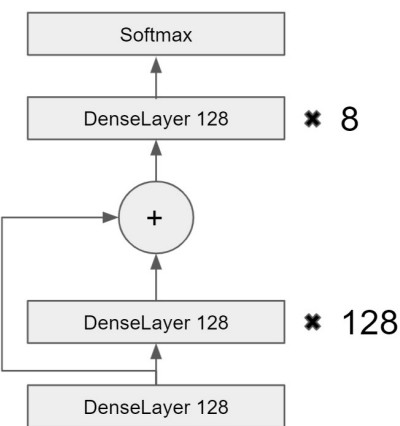

(a) The multi-layer perceptron architecture used for the reproduction of the experiment on skip connections by Alain et al. (2020). The architecture deviates slightly from the original by Alain et al. (2020) to make the model easier to train.

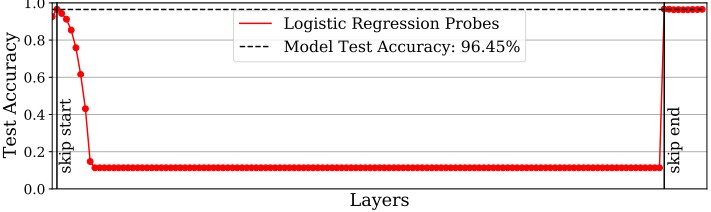

(b) The performance of the logistic regression probes in the order of the forward pass. Note the stark difference in behavior between the first part (encapsulated by the skip connection) and the second part of the network.

Figure 2: The images show the basic neural architecture with skip connection (a) and the probe performances of each layer (b). The setup is designed to provoke the network to "skip" the 128 layers by learning some identity-mapping analog. By observing the probe performances, this behavior can be observed. After the initial layer, the performance degrades until reaching chance level. The probe performances recover as soon as the skip connection is added to the layers again. We will observe similar behavior on convolutional neural networks over the course of this work.

Since we are interested in the information processing of (partial) networks, tools that easily allow the comparison of learned feature representation are required. Several works (Raghu et al., 2017;

Morcos et al., 2018; Feng et al., 2020) proposed such tools to gauge the similarity of the features learned by two models having the same architectures but trained with different initialization. The similarity is calculated by taking the hidden state activations from each model in the form of a feature matrix and finding a measure of correlation between them. The method proposed by Wang et al. (2018) uses a subspace match model to find the similarity in terms of maximum and simple matches. Raghu et al. (2017) uses Canonical Correlation Analysis (CCA) proposed in Golub & Zha (1995) coupled with Singular Vector Decomposition (SVD) to compare the representations presented by two neural networks in a process which is invariant to invertible linear transformations. The technique by Morcos et al. (2018) further improves this method by giving more weight to those intermediate CCA vectors which are more important for the representations. Another metric proposed by Feng et al. (2020) called the Transferred Discrepancy (TD) focuses on the practical usage of the representations by stating their similarity based on their performance in downstream tasks.

Another approach to compare layer activations, that has gained popularity over the last years is Centered Kernel Alignment (CKA, Cortes et al., 2012), which we will use for this work. It is introduced as a normalized version of the Hilbert-Schmidt Independence Criterion (HSIC, Gretton et al., 2005), and when used with linear kernels, it is equivalent to the RV coefficient Robert & Escoufier (1976). In this form it has been applied successfully in recent works as an efficient alternative to SVCCA and other methods (Nguyen et al., 2021; Kornblith et al., 2019), majorly because it requires less number of data points (Kornblith et al., 2019).

## 3 DISTRIBUTION OF INFORMATION PROCESSING IN NETWORKS WITH SKIP-CONNECTIONS

In this section, we will investigate the first scenario we identified in which a network with a skip connection will reliably choose not to utilize the layers encapsulated by a skip connection. Skipping of layers could be described as an autogenous pruning technique, where the model decides during training to not utilize certain layers. Therefore, we hypothesize that layers that would be unproductive are likely to be skipped if the network is given the opportunity. We find that it is possible to guarantee a convolutional layer will not be able to contribute to the quality of the solution by utilizing the knowledge about the relation of receptive field and input resolution presented by Richter et al. (2021b). The authors show experimentally that convolutional layers can only enhance the quality of the solution if the receptive field of the layer's input is smaller than the input image. Simply speaking, if the layer is unable to integrate novel information into a feature map position by convolving the kernel over the input, the layer will not improve the quality of the intermediate solution.

In case of a simple sequential architecture neural network like VGG16 the layers pass the solved problem from layer to layer while not enhancing the intermediate solution quality, as Fig. 3 (a) exemplifies. The observed effect is caused by training the model on Cifar10 using the native resolution of $32 \times 32$ pixels. When repeating this experiment on DenseNet18, we can see that the probe performance no longer stagnates. Instead, the final dense-block of the model is skipped entirely, which is apparent when looking at the decaying probe performance in Fig. 3 (b). A similar effect can be observed when training ResNet34 in the same scenario. In this case, unproductive residual blocks are skipped. Since the input of a residual block is added to its output, the performance recovers after each building block and thus recovering the intermediate solution quality. This is reflected by a zig-zag-pattern in the LRP.

Based on these results, we can see that unproductive layers are behaving differently in architectures where these layers can be bypassed. Instead of learning to functionally emulate a pass-through layer, the layers learn a representation which is functionally equivalent to an identity mapping. With functionally equivalent, we refer to the behavior of the layers being similar to a pass-through layer or an identity mapping regarding the predictive performance. We find that skipped layers do in-fact not learn an identity mapping. Instead, these layers learn a *harmless* non-zero representation that does not hurt the intermediate solution quality when added to the feature map containing a good solution. This makes it also impossible to remove these skipped layers in a primitive pruning step without retraining the model, since the changes in the data caused by the unproductive layers are

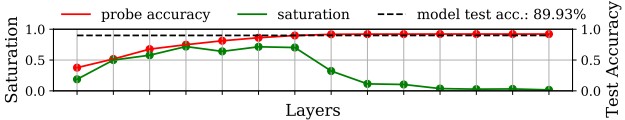

(a) VGG16 tail layers maintain the quality of the intermediate solution.

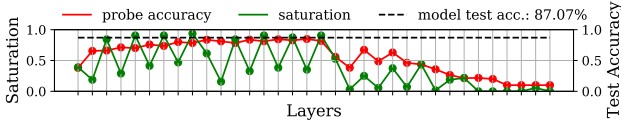

(b) The tail of DenseNet18 shows a decay in probe performance, indicating that the last DenseBlock is skipped entirely (Alain et al., 2020).

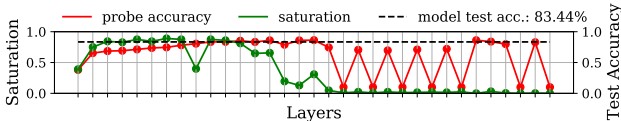

(c) ResNet34 skips most residual blocks in the tail, which is apparent by the zig-zag pattern in probe performances caused by the starts and end of skip-connections (Alain et al., 2020).

Figure 3: Depending on the neural architecture, tail patterns may deviate in their appearance in probe performance. In sequential architectures (a) the layers maintain the quality of the intermediate solution. If shortcut connections exist in the architecture, layers may be *skipped*. Skipped layers are apparent by their decaying probe performance Alain et al. (2020). This is apparent on DenseNet18 (b) and ResNet34 (b) where a single DenseBlock and multiple ResidualBlocks are skipped respectively. All models are trained on Cifar10 at native resolution.

still expected by the following layers. Nevertheless, removing the skipped layers and retraining the pruned model will lead to a more efficient network (Richter et al., 2021b).

## 4 DISTRIBUTION OF INFORMATION PROCESSING IN MULTI-PATH ARCHITECTURES

We proceed to investigate networks with parallel sequences of layers. To make the results of this investigation easier to visualize, we use a simple CNN architecture that is composed of building blocks that contain two distinct pathways. The basic template for this architecture can be seen in Fig. 4. The architecture itself consists of four stages consisting of $k$ building blocks each. The filter size is doubled from stage to stage, while the first layers to process the data in every stage downsample the filter map with convolutions with stride size 2. The downsampling and the increases in filter sizes over multiple stages are a common feature in modern neural architectures such as ResNet He et al. (2016) or EfficientNet (Tan & Le (2019)). The individual building blocks have a two-pathway layout, see Fig. 4 (b). Each pathway is a sequence of layers using the same kernel size. The number of layers and kernel sizes in each pathway are varied throughout the experiments and will be explicitly mentioned in the respective experiment. The pathways are reunited using an element-wise addition. We decided to utilize element-wise addition instead of concatenation, since it avoids the otherwise necessary $1 \times 1$ convolution for dimension reduction after each building block. From additional experiments, we find that the $1 \times 1$ convolutions also do not impact the results of the experiments substantially. We refer to this multipath architectures as $\text{MPNet}_K(n_1 : s_1 \times s_1, n_2 : s_2 \times s_2)$, where $K$ is the number of building blocks, $n_1$ and $n_2$ refer to the number of

layers for the first and second pathway and $k_1$ and $k_2$ refer to the kernel sizes used in the first and second pathway respectively. We refer to the sequence of layers composed by only the pathways that feature the smaller kernel size as $p_{l,min}$ and to the path with the larger kernel sizes as $p_{l,max}$. While these pathways are intertwined in the architecture, we visualize these two pathways as separate sequences, which makes the visualizations more digestible.

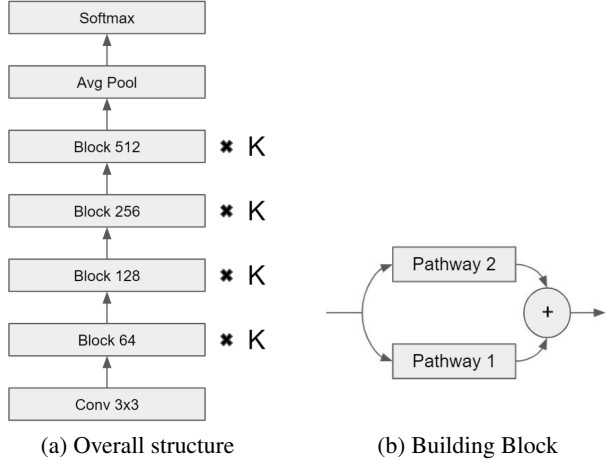

(a) Overall structure          (b) Building Block

Figure 4: The architecture used for the experiments in this section. Each stage consists of $k$ blocks and has double the filters of the previous stage. The first layer of each stage is a downsampling layer with a stride size of 2. Each building block consists of two pathways, which resemble sequences of convolutional layers.

Training is conducted on the datasets of ImageNette and Cifar10. We choose ImageNette over the full ImageNet dataset, since the large number of experiments is necessary to make the results presented in this work quantifiable. Since LRPs are very resource and time intensive to train on large datasets, we make this concession to practicality. Training is conducted using the same preprocessing as He et al. (2016), training is conducted for 30 epochs using stochastic gradient descent with a learning rate of 0.1. The learning rate is reduced by a factor of 0.1 every 10 epochs.

### 4.1 DOMINANT AND NON-DOMINANT PATHWAYS IN HETEROGENEOUS BUILDING BLOCKS

We first investigate scenarios where the pathways behave substantially different when measured with LRP. We find that this can be achieved by increasing the heterogeneity between the different pathways. To exemplify this, a building block is used with a $3 \times 3$ pathway with 4 layers and a $7 \times 7$ pathway with a single layer. These two pathways differ in three important properties. The number of layers, resulting in a strong difference in capacity, receptive field and gradient flow. We train these models on ImageNette with an $250 \times 250$ input resolution. When using a single building block per stage, we can see in Fig. 5a that now the second building block is only utilizing the short pathway. When doubling the number of building blocks per stage, we observe in Fig. 5b that this behavior has spread to all building blocks. While this indicates that $p_{l,min}$ is utilized while layers of $p_{l,max}$ are skipped, we were unable to confirm this experimentally. When removing the *skipped* layers, we find that the performance decreases by 2.16%-points on average, which indicates that's these paths are still not entirely irrelevant for the inference process. For this reason, we will not refer to these paths as *skipped*. Instead, we will refer to the path as *dominant* when they advance the intermediate solution quality directly and *non-dominant* when their contribution is not directly measurable by logistic regression probes.

### 4.1.1 PARTIAL VANISHING GRADIENT IN NON-DOMINANT PATHWAYS

So far, we observe two distinct causes for degradation in LRP performance. First, the degradation caused by an excessively large receptive field in section 3, second the dominant/non-dominant behavior observed in section 4.1 caused by heterogeneity in the pathways. The pathways of the previously tested models in section 4.1 differ in depth and receptive field size. We hypothesize that the

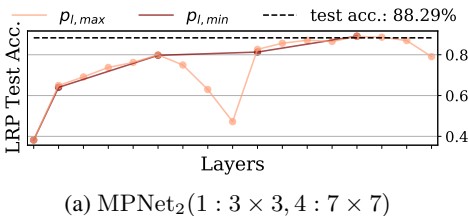

(a) $\mathrm{MPNet}_2(1:3\times3,4:7\times7)$

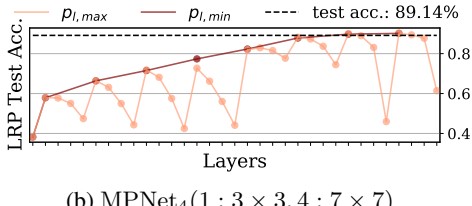

(b) $\mathrm{MPNet}_4(1:3\times3,4:7\times7)$

Figure 5: When training MPNet with pathways of different depth, a deeper pathway is skipped for a building block (a), which is apparent by the deteriorating probe performance. This effect occurs on all building blocks when the depth is increased. The models are trained on Cifar10.

observed behavior is caused by one of those two factors. We test this hypothesis by using a building

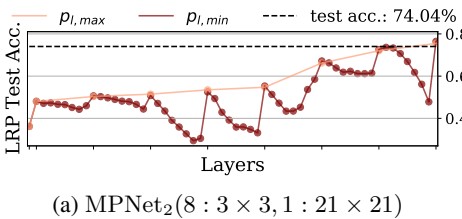

(a) $\mathrm{MPNet}_2(8:3\times3,1:21\times21)$

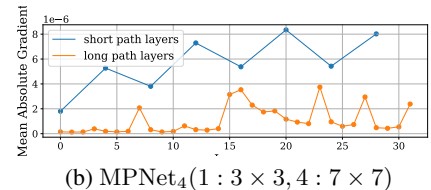

(b) $\mathrm{MPNet}_4(1:3\times3,4:7\times7)$

Figure 6: Different from tail patterns, the shallower pathway is dominating the deeper pathway even if their receptive field sizes are identical (a). We attribute the dominating-behavior to a vanishing gradient in the deeper pathway, evident in the overall lower size of the gradients (b).

block with a pathway using $8\,3\times3$ layers and a second pathway using a single $21\times21$ convolutional layer. The result of this unusual setup is that the single layer on the $p_{l,min}$-pathway will increase the receptive field more than the stack of $3\times3$ in the other pathway. If the receptive field is responsible for this behavior, we should be able to observe a skipping-behavior in the shorter path. As we can see in Fig. 6a, this is not the case. Instead, the layers still appear to be dominant for the most part. A better explanation for this behavior is provided by analyzing the accumulated gradients of a neural architecture with a clear dominant/non-dominant behavior lime $\mathrm{MPNet}_4(1:3\times3,4:7\times7)$ trained on ImageNette using an $160\times160$ input resolution in Fig. 6b. This is done by computing the average over the absolute mean of all gradients, which gives us an idea on the *size* of the gradient that is propagated back through a given layer. We can clearly see that the gradients of the long, non-dominant pathway are strictly smaller than the gradient of the dominant pathway. We interpret this as a partial vanishing gradient problem, where the gradient flow is following the *path of least resistance*, implicitly preferring shorter pathways to longer ones, where the signal may be more diluted by the number of layers the gradient has to pass through.

## 4.2 Coexistence of homogeneous parallel Pathways in Multipath Architectures

We continue to analyze scenarios in which both pathways are less heterogeneous. For this reason, we train a $\mathrm{MPNet}_2(2:3\times3,2:7\times7)$ on Cifar10. The equal number of layers in both pathways ensures that the gradient travels through a similar amount of layers in both pathways. We chose the kernel sizes to make both pathways slightly heterogeneous, similarly to the different pathways in the Inception architectures. We additionally train $\mathrm{MPNet}_2(1:3\times3,2:7\times7)$, which features one less $3\times3$ convolution to increase the heterogeneity of the pathways. We train both models on Cifar10 and train LRP on each layer of the architecture. From the results in Fig. 7 we observe that the probe performance of the two pathways $p_{l,min}$ and $p_{l,max}$ are almost indistinguishable. We interpret this as a coexisting behavior, these pathways do not necessarily extract different features to achieve the depicted increase in LRP accuracy. We will see further evidence of this in the next section.

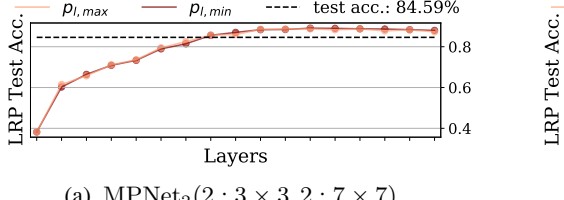 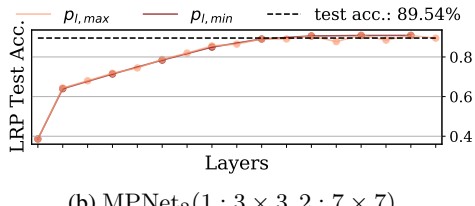

(a) $\mathrm{MPNet}_2(2 : 3 \times 3, 2 : 7 \times 7)$       (b) $\mathrm{MPNet}_2(1 : 3 \times 3, 2 : 7 \times 7)$

Figure 7: The MPNet architectures with only slightly different pathways in each building block show a coexisting behavior, which is indicated by the probe performances of the $3 \times 3$-pathway $p_{l,min}$ and the $7 \times 7$ pathway $p_{l,max}$ increasing at a similar rate.

### 4.2.1 CORRELATION ANALYSIS IN MULTIPATH ARCHITECTURES

To provide additional insights and also a different perspective on the division of information in multipath networks, we utilize the CKA as a metric to quantify correlations. While our main analysis tool, LRP, measures the quality of the information being carried in terms of accuracy, the CKA can provide the correlation measurement between the latent representations of layers. We compute the CKA for all the layers (for both the paths) with the final output of the network. This is calculated in a similar way as the LRP accuracy (see section 2.2). The resulting graphs can be seen in the Fig. 8.

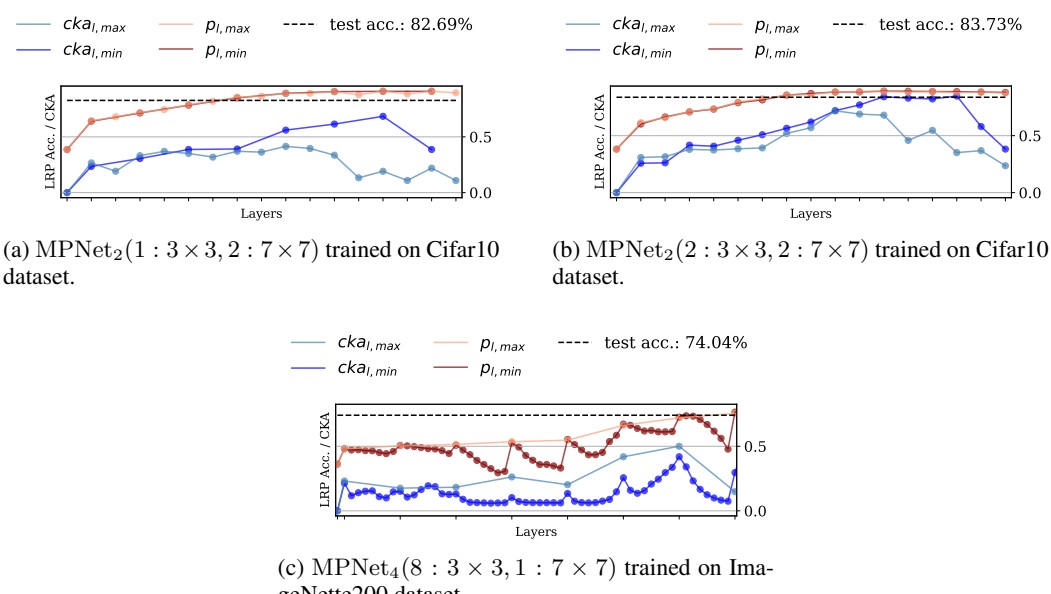

(a) $\mathrm{MPNet}_2(1 : 3 \times 3, 2 : 7 \times 7)$ trained on Cifar10 dataset.

(b) $\mathrm{MPNet}_2(2 : 3 \times 3, 2 : 7 \times 7)$ trained on Cifar10 dataset.

(c) $\mathrm{MPNet}_4(8 : 3 \times 3, 1 : 7 \times 7)$ trained on ImageNette200 dataset.

Figure 8: Comparison of the representational similarity score given by the Centered Kernel Alignment (Cortes et al., 2012; Kornblith et al., 2019) and the probe performance of MPNet architectures with different sets of pathways in each building block. The CKA value is calculated to quantify each layer's representational similarity with the final output of the network. The $min$ and $max$ subscript refer to the path with the smaller and larger kernel sizes. The CKA values are in accordance with the probe performance, indicating that the shorter paths and the longer paths are extracting more similar features in early layers of the network when compared to later layers.

We observe that initially both the shorter and the longer paths have relatively similar internal representations because the plots of CKA for these paths are close to each other. In the later stage, we can see that the CKA values are significantly different from each other, which signifies that shorter paths and longer paths are learning different representations in the later stage of the network. Another

trend that we observe is, that the shorter path exhibits higher internal representation correlation with that of the final output of the neural network than the longer path. This is substantial as the CKA values of the shorter path are higher throughout compared to the CKA values of the longer path.

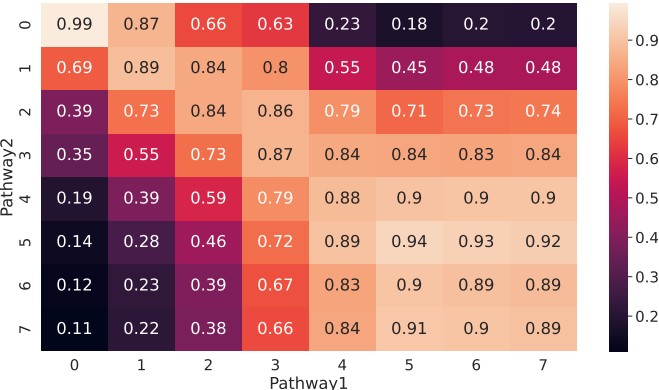

Figure 9: $\text{MPNet}_2(8 : 3\times 3, 8 : 3\times 3)$ trained on ImageNette dataset showing the correlation among 8 layers deep parallel pathways. We observe that layers of the two pathways have high redundancy with their respective counterparts in the other pathway, indicating a parameter inefficiency.

.

Additionally, we are interested in learning how much work is shared when pathways are very deep and identical in their structure. We test this by analyzing the last building block of $\text{MPNet}_2(8 : 3 \times 3, 8 : 3 \times 3)$ trained on ImageNette. Since both pathways are identical, we can compare the layers in a building block directly with their counterparts from the other pathway. This is different compared to the previous experiment, where we measured CKA similarity of all layers with the final output of the network. When looking at the CKA similarity heatmap of a building block in Fig. 9, we can observe that layers in the first pathway show a high similarity with their respective counterparts in the other pathway, implying a high degree of redundancy in the extracted features. This could be considered a parameter inefficiency.

## 5 DISCUSSION AND CONCLUSION

In this work, we analyzed the distribution of the inference process in multipath architectures. We first investigated sequential models with skip connections like ResNet and DenseNet. For these architectures, we find that layers are skipped when they are part of the tail pattern (see the works of Richter et al. (2021a) and Richter et al. (2021b)), which can be predicted by applying the border-layer rule of Richter et al. (2021b). We then moved on to architectures with multiple pathways in each building block. Our analysis suggests, that only the pathways with fewer layers are utilized if the parallel pathways differ strongly in depth. We were able to attribute this to a localized vanishing gradient problem experimentally. When the pathways are homogeneous, the behavior of the pathways changes to a coexisting behavior, where both pathways improve the intermediate solution quality at a similar pace. We find that slight differences in the pathways result in increasingly distinct extracted features in later layers. However, if the pathways are identical, also the extracted features become identical according to our CKA analysis.

It can be argued that the parameters of the layers in multipath architectures are not utilized fully in the heterogeneous and homogeneous case. In one scenario because of the redundancy in extracted features, and in the other case because of layers not directly advancing the intermediate solution quality. Following the line of thought, architectures with parallel pathways may be prone to inefficiencies. This may also explain why most efficient state-of-the-art architectures in recent years like ResNet, MobileNet and EfficientNet only feature skip connection, which do not suffer from this problem.

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
