# OpenReview forum: "Go with the Flow: the distribution of information processing in multi-path networks"
_ICLR.cc/2022/Conference — ICLR 2022 Submitted_

### Official Review · Reviewer_PRea · 2021-10-31

**Correctness:** 3
**Technical Novelty And Significance:** 2
**Empirical Novelty And Significance:** 2
**Recommendation:** 5
**Confidence:** 3

**Main Review:**

Strengths:
1) The motivation is interesting that we can design better neural networks without training. Some of the experiments are interesting that they find latter layers of ResNet and DenseNet can be skipped for model efficiency and without performance drop.


Weakness:
1) It seems to me that the Section 3 is redundant since the message is the same as the previous paper in the citation. [1]

2) I think the Section 4 should be ordered better. It is not necessary to discuss the heterogeneous and homogeneous cases since the paper mainly cares about the receptive field size and the depth of paths. Then you just analyze the blocks with different settings of receptive field size and the depth. And the results that shorter paths are dominant and the longer paths that suffers from gradient vanishing is not surprising.

3) There're multiple typos and errors in the paper. For example, at the end of page 5, the $s_1$ and $s_2$ should be $k_1$ and $k_2$. I think the citation [2] in Figures 3 is wrong. For Section 4.1, it says that a building block is used with a 3 * 3 pathway with 4 layers and a 7 * 7 pathway with a single layer, however, in Figure 5, it seems the pathway with 7 * 7 kernels has 4 layers instead.

4) For the homogenous case in the paper, can we have the results that only on pathway is used?


References:
[1] Mats L. Richter, Julius Schoning, and Ulf Krumnack. Should you go deeper? optimizing con-volutional neural network architectures without training by receptive field analysis.

[2] Guillaume Alain, Maxime Chevalier-Boisvert, Frederic Osterrath, and Remi Piche-Taillefer. Deep- drummer : Generating drum loops using deep learning and a human in the loop.

**Summary Of The Paper:**

This paper analyzes the distribution of information processing in multi-path networks (including skip-connection models such as ResNet and DenseNet). They apply logistic regression on the hidden layers, namely logistic regression probes, to track the progress of the intermediate solution quality, and they analyze in which condition (depth of the path and receptive field size) the neural network prefers to skip the paths. They also measure the CKA similarity of the hidden representations learned by the multi-path model when the paths are homogeneous. They claim that with their analysis, later layers in ResNet and DenseNet can be skipped as pruning due to the unproductive layers. In addition, they find that for multi-path model, the shorter path is dominant when the depth of the paths are very different, and when the pathways are homogeneous, the behavior of the pathways changes to a coexisting behavior.

**Summary Of The Review:**

Although the motivation of the paper is interesting, the analysis results of the paper is weak and not that surprising. In addition, part of the experiments are redundant since it is mentioned in the previous paper. There's little new insights or takeaways from the analysis. As a result, I recommend to reject the paper.

---

### Official Review · Reviewer_ac9X · 2021-11-01

**Correctness:** 3
**Technical Novelty And Significance:** 1
**Empirical Novelty And Significance:** 2
**Recommendation:** 3
**Confidence:** 5

**Main Review:**

The paper has following strengths:
1.	Studying multi-path networks with respect to receptive field and depth is an interesting problem.
2.	The paper is well-written and motivation is very good.

The paper has following weaknesses:
1.	Unfortunately, the conclusions that the authors could draw from their study are quite limited and, in most cases, completely obvious (or already established by previous papers). For instance, their first contribution in introduction: input size vs. receptive field size (as cited by the authors themselves) was already published in Richter et al. 2021a [R1] and 2021b [R2]. Other discussions about layers getting completely skipped in ResNets and variants is extremely well-established at this point (see also the third point below). The second contribution in introduction on preferring shorter pathway over longer pathway due to vanishing gradients: At first, this is also obvious since many papers (starting with the ResNet paper [R3] itself) have established that consecutive convolution layers without skip connections will result in diminishing gradients. However, these gradient properties can be modified if each branch in the multipath network further had skip connections within individual branches too. In such more complex cases, the conclusions of this study will not hold. The third contribution is also completely obvious, and the reviewer is not sure what else one can expect. If both branches in the multipath network are identical, then of course they will move the solution at a similar pace and will have high redundancy.
2.	Another concerning issue is that it is not immediately clear if the findings in this paper will apply to more complex real-world problems or how to practically use the knowledge from this paper. The authors have conducted all experiments on CIFAR-10 and ImageNette datasets. The reviewer understands that doing such studies require a lot of ablations and time-/resource-consuming experiments. However, questions like “would a network more effectively utilize a shallower, higher receptive field branch over a deeper, low receptive field branch?” should also depend on the complexity of the problem itself. CIFAR-10 is a toy dataset, and while ImageNette has a larger input size, the creators of that dataset still classify it as a very simple dataset (see: “Imagenette is a subset of 10 easily classified classes from Imagenet” on https://github.com/fastai/imagenette). So, the reviewer is left wondering if this analysis will hold for more complex datasets (say CIFAR-100, tiny imagenet, etc.)? In the current form, the study seems inconclusive.
3.	The reason why many of the conclusions appear obvious to the reviewer (particularly anything related to deeper pathways not being used in DNNs over shorter pathways when skip connections are used) is because of the existing literature. Two very important papers in this direction have not been discussed. Veit et al. already established long ago that ResNets behave like an ensemble of shallower networks [R4]. More recently, Bhardwaj et al. theoretically and empirically discussed the gradient flow properties through network topologies with skip connections [R5]. They derive theory on DenseNet-type models but show empirical results for MobileNets/ResNets as well. These works must be discussed in detail.

[R1] Mats L. Richter, Wolf Byttner, Ulf Krumnack, Ludwig Schallner, and Justin Shenk. (input) size matters for convolutional neural network classifiers. In Igor Farkaˇs, Paolo Masulli, Sebastian Otte, and Stefan Wermter (eds.), Artificial Neural Networks and Machine Learning (ICANN), volume 12892 of Lecture Notes in Computer Science, pp. 133–144. Springer, Cham, 2021a.

[R2] Mats L. Richter, Julius Sch¨oning, and Ulf Krumnack. Should you go deeper? optimizing convolutional neural network architectures without training by receptive field analysis. CoRR, abs/2106.12307, 2021b. URL https://arxiv.org/abs/2106.12307.

[R3] Kaiming He, Xiangyu Zhang, Shaoqing Ren, and Jian Sun. Deep residual learning for image recognition. In 2016 IEEE Conference on Computer Vision and Pattern Recognition, CVPR 2016, Las Vegas, NV, USA, June 27-30, 2016, pp. 770–778. IEEE Computer Society, 2016. doi: 10.1109/CVPR.2016.90. URL https://doi.org/10.1109/CVPR.2016.90.

[R4] Veit, Andreas, Michael J. Wilber, and Serge Belongie. "Residual networks behave like ensembles of relatively shallow networks." Advances in neural information processing systems 29 (2016): 550-558.

[R5] Bhardwaj, Kartikeya, Guihong Li, and Radu Marculescu. "How does topology influence gradient propagation and model performance of deep networks with DenseNet-type skip connections?." Proceedings of the IEEE/CVF Conference on Computer Vision and Pattern Recognition. 2021.


**Summary Of The Paper:**

The paper studies information processing in (1) networks with skip connection like ResNets and DenseNets, and (2) multi-path networks where the model has multiple branches with homogeneous or heterogeneous layers. The study focuses on problems like: (a) does learning depend on receptive field and/or depth of various branches, (b) how features differ in representation at different stages of the network

**Summary Of The Review:**

The paper seems like a good first attempt towards an important and difficult problem. However, most of the conclusions seemed obvious or inconclusive (e.g., since the paper does not account for “how difficult a given problem is”, it is not easy to conclude how much receptive field size we really need over depth, etc.). More comprehensive evaluation on difficult datasets (not necessarily large datasets) is needed for a better contribution. Note that, most statements made in the paper themselves are correct and well-supported. The reviewer has concerns about the claims being obvious.

---

### Official Review · Reviewer_Hchc · 2021-11-01

**Correctness:** 3
**Technical Novelty And Significance:** 2
**Empirical Novelty And Significance:** 2
**Recommendation:** 3
**Confidence:** 3

**Main Review:**

[Pros]

* The paper proposes interesting new experimental framework to analyse the behaviour of learned representations in popular deep CNNs. The idea to concurrently use LRPs, representation saturation (% of dimensions required after PCA) and techniques such as Centered Kernel Alignment (CKA) to analyse post-hoc behaviour is valid

* The conclusion that minor heterogeneity in network pathways leads to distinct laterr feature deeper in the network is insightful and can potentially lead to new network designs in the future

[Cons]

* The bulk of the experiments was performed on the hypothetical network shown in Figure 4 that was constructed to feasibly allow analysis of the effect of different paths. However, it seems as if the results are very specific to this network. There is no mention of how these findings generalise to multi-path networks in the literature. Further more, if heterogeneity/homogeneity of paths is important for representations and performance, I would have liked to see these measured in relation to popular network to better understand why a ResNet might perform better than another.

* Section 3 and Figure 3 are very similar to Figure 2 from Richter et al. 2021b. The main novelty of this work appears to simply extend the experimental procedure of Richter to morre complex network architectures that go beyond sequential networks (VGG style)

* I am missing what the overall conclusion of this paper is. The experiments aim to demonstrate the redundancy of featurres learned or the fact certain kernels do not learn anything meaningful when comparing short versus long-paths. However, there are no real recommendations to take these results onboard. Could these metrics be used in a data-driven way through neural architecture search or to prune layers iteratively during training? How might one use the observations of this paper when designing new methods? I feel that the observations do not lead to significant insights that may help advance the field.

[Other]

* I would have liked to see the results are different stages of training. Further information on the dynamics of gradient updates, feature similarity and representation saturation in different multi-path networks might help better understand the results and may point to new training schemes. For instance https://proceedings.neurips.cc/paper/2017/file/dc6a7e655d7e5840e66733e9ee67cc69-Paper.pdf showed bottom-up convergence, leading to the idea of Freeze Training.



**Summary Of The Paper:**

This is an experimental paper that seeks out to investigate information processing in multi-path networks i.e. networks such as ResNet, EfficientNet, Inception-style. The goal was to investigate how different pathways process information in such networks in order to better understand learned representations to inform new network architectures in the future. The authors used logistic regression probes (LRP) and representation saturation as metrics to power their analysis. Through the analysis of a network composed of many multi-path networks with differing number of layers or different receptive fields, the authors demonstrate that shorter pathways often dominate longer pathways. Further, if pathways are slightly difference, distinct features will be promoted in later layers.

**Summary Of The Review:**

The paper is interesting and uses a suite of analytical tools to evaluate the representations learned in different multi-path network settings.

However, in the current format, it is hard to see how the results of the experiments generalise to networks used in the literature, and which aren't directly consistent with the network used for experiments. Furthermore, the analysis is very similar to Richter et al. and the main novelty is its application to multi-path network.

For this reason, I am currently recommending Rejection

---

### Decision · Program_Chairs · 2022-01-20

**Decision:**

Reject

**Comment:**

The paper analyzes the flow of information in convolutional neural networks with parallel pathways by using logistic regression probes and similarity indices.
Following the analysis, the authors concluded that:
- pathways of similar size have similar contributions to learning and have high redundancy
- shorter pathways directly improve solution quality to a greater extent than longer pathways
- pathways of different lengths also lead to grater variety amid features in the 'downstream' layers of the network

The novelty in this type of analysis and its thoroughness was appreciated by the reviewers. The insight about the benefits of pathways of different length is also valuable; although there is a sense in the community that long pathways without skip connections bring diminishing returns, as pointed by reviewer ac9X, there is still some benefit in quantifying it through such an analysis and establishing the correct mix of long/short pathways. [note: This paper is not there yet, but has the potential.]

On the other hand, the experiments were performed on a single network, a network selected such that this instrumentation is possible, so there is the issue of whether these conclusions generalize, as pointed out by reviewer Hchc.

There were other comments, in terms of structure, errors and typos, as pointed out by reviewer PRea.

The authors have not responded to the comments, nor updated their manuscript.
In its current form, the paper is not ready for acceptance.